# COVID-19 the Gateway for Future Learning: The Impact of Online Teaching on the Future Learning Environment

Badr A. Alharbi [1], Usama M. Ibrahem [2,3,*], Mahmoud A. Moussa [3], Shimaa M. Abdelwahab [4] and Hanan M. Diab [2]

1   Faculty of Education, University of Ha'il, Ha'il 81451, Saudi Arabia
2   Computer Science Department, Community College, University of Ha'il, Ha'il 81451, Saudi Arabia
3   Faculty of Education, Suez Canal University, Ismaïlia 41522, Egypt
4   Faculty of Education, Mansoura University, Mansoura 35516, Egypt
*   Correspondence: u.abdelsalam@uoh.edu.sa

**Abstract:** The COVID-19 virus has altered the nature of education. These modifications may be reversed once universities reopen. Nevertheless, a few of these modifications afford novel options to match pre-COVID-19 suggestions. This study's purpose is to study staff members' perceptions of online teaching during COVID-19, describe future projections regarding teaching, and identify the drivers of change in the future learning environment. The study community represents faculty staff in Saudi universities. The sample consisted of 127 faculty staff in nine Saudi universities. Participants had positive perceptions of the usage of e-learning platforms during COVID-19 according to data analysis (though negative experiences existed). Future research must focus on the subtle challenges of aligning theoretical and methodological designs to appropriately analyze the phenomenon under inquiry while contributing to a well-executed body of research in the field of educational technology. Future study is required to determine how teachers perceive information and communications technology (ICT) trading strategies in the light of COVID-19.

**Keywords:** COVID-19; online teaching; cloud computing environment (CCE); future learning; emergency remote teaching; learning crisis

## 1. Introduction

The COVID-19 global pandemic began in 2020. It severely impacted society, posing global education-related challenges. Many governments ordered institutions to cease face-to-face instruction, requiring students to switch to online teaching and virtual education very quickly [1]. As a positive response to COVID-19, digital technologies were utilized for this purpose.

Massive technological advances in the world demand a paradigm shift in how we approach our educational goals. Many educational institutions have adopted ICT tools such as laptops and projectors. Beside, Today's students appreciate technology-integrated learning [2,3].

The COVID-19 pandemic measures allowed a critical gateway for educational institutions to develop symbiotic plans using pedagogical, technological, economic, social, and geopolitical elements that could effectively propel the industry into the future while contending with complexity and ambiguity [1]. COVID-19 was a catalyst for change and a new way of doing things.

Although face-to-face instruction will likely return to institutions, most COVID-19 countermeasures will prevail. The expansion of online learning in tertiary education will further accelerate, and schools will organize themselves more systematically to pursue the aspects of technology-based learning as a result of its benefits [4].

Crises have always had an impact on education [5]. This can be countered only by employing atypical technologies in multiple forms and by being prepared.

Technology is the future of education, expedites, and enhances learning. Artificial intelligence and smart applications will be a tool that aids us in our quest for a new era of enlightenment, transforming education into something more immersive and engaging [1,6].

This research holds high significance in troubled times, enabling a new perspective for educators, policymakers, and parents with regard to how to effectively manage remote teaching. Thus, future research needs to examine the consequences of expanding and embedding digital technologies and media in education systems and practices in future learning. The study's primary question is "In light of teaching during COVID-19", what are staff members' future opinions of remote teaching.

## 2. Literature Review

### 2.1. COVID-19 and Distance Learning: Challenges and Opportunities

For all higher education stakeholders, the pandemic poses a unique challenge. All higher education institutions were instructed to use emergency remote teaching (ERT) modes to quickly ensure sustainable education while reducing transmission risk. ERT was a temporary shift in instructional delivery from face-to-face to online delivery [7]. However, readiness-related concerns were raised, especially whether teachers, administrators, students, and their parents were prepared for this shift.

The COVID-19 pandemic has made extraordinary demands on learners, faculty staff, and administrators. The lack of Information and communication technology (ICT) related skills created additional barriers to teaching and remote learning [8], besides the challenge of assessing students' skills and development [9]. The need for prompt actions negated the careful considerations for a smooth online learning experience [2]. There were obstacles concerning conducting of tests and the evaluation of e-exams, in addition to teachers' aversion to migration from face to face (F2F) to online during the pandemic, e.g., preparedness, confidence, support from the institution, etc. [10].

Adedoyin and Soykan summarized the challenges of this digital transformation during COVID-19 for higher education institutions: (a) the compatibility gap of the online application of some disciplines (remote laboratories limitations), (b) outdated technology and accessibility problems, and (c) regulating and avoiding cheating complications [11].

### 2.2. Beyond Learning during COVID-19

COVID-19 is an once-in-a-lifetime chance for true change. Due to the pandemic's global scale, it presented an opportunity for educators and students to rethink education. Second, educators all over the world showed that they could adapt as a group. Third, it supports smart devices and appropriate network infrastructure [12].

This unique situation enabled a rich educational experience. It has spurred the introduction of many innovative, technology-driven approaches to education with potential longer-term benefits for education [3,13]. ERT tools enabled teachers to deliver educational content within the complete context students' curricula [1]. Teachers should be trained by pedagogical professionals in the use of specialized ERT instruments such as online course delivery platforms, class size control, and so on. [14,15].

### 2.3. Different Future Students, Different Future Needs

A recent poll revealed that 75% of educators believe digital content will replace textbooks by 2026. Choosing which innovations to bring into the classroom is somewhat of a challenge [2,16]. Students must use many cognitive, motivational, behavioral, and contextual components to be successful in digital learning. Students will have additional learning opportunities at various times and locations. Remote, self-paced learning is facilitated with e-learning tools [17,18]. Though tools to assess many of the intended outcomes are not present, alternative assessment techniques for different objectives could help [19,20].

*2.4. The Future Learning*

As the world embraces technological futures, so will the way of teaching need to keep up with the expanding demands of the 21st century. Though not completely novel, student-learning spaces will eventually supplant the traditional classroom, making students collaborators in their learning [20,21]. Future students will have much autonomy in their learning, and mentorship will become critical to their success [22]. Curricula will make way for abilities that exclusively require human knowledge and face-to-face interaction, as technology allows for greater efficiency in certain sectors. Courses will place a premium on "on-the-job" experience [17,22].

## 3. Research Question

This research aims to collect educators' thoughts on using online learning resources during the recent COVID-19 pandemic. Also, during the COVID-19 pandemic, it is important to examine teachers' difficulties adjusting to the online teaching and learning process. Furthermore, for the purpose of argument, the question is how effective education was during the COVID-19 epidemic and how future educators would view the profession.

The study addressed the following research question: what are the future perceptions of staff members about remote teaching in light of teaching during the COVID-19 pandemic?

## 4. Methodology
### *4.1. Participants*

The study community represents faculty staff in Saudi universities, 127 faculty staff in nine Saudi universities in Hail, Umm Al-Qura, Qassim, Imam Abdul Rahman bin Faisal, Imam Muhammad bin Saud Islamic, King Faisal, King Saud, King Abdulaziz, and Al Baha Universities, as in Table 1:

**Table 1.** Demographic information for the study sample: University, Academic Rank, gender, and years of experience.

| No. | Demographic Information | | *n* | % of Total |
|---|---|---|---|---|
| 1 | University | Hail University | 25 | 19.6 |
| | | Umm Al Qura University | 20 | 15.7 |
| | | Imam Muhammad Bin Saud Islamic University | 11 | 8.6 |
| | | Al Qussaim university | 15 | 11.8 |
| | | King Saud University | 12 | 9.4 |
| | | King Abdulaziz University | 11 | 8.6 |
| | | King Faisal University | 8 | 6.2 |
| | | Al Baha university | 13 | 10.2 |
| | | Imam Abdul Rahman bin Faisal University | 12 | 9.4 |
| 2 | Gender | Male | 51 | 40.2 |
| | | Female | 76 | 59.8 |
| 3 | Academic Rank | Professor | 20 | 15.7 |
| | | Associate Professor | 26 | 20.4 |
| | | Assistant Professor | 81 | 63. |
| 4 | Years of Experience | Less than 5 years | 62 | 48.8 |
| | | From 5 years to less than 10 years | 11 | 8.6 |
| | | From 10 years to less than 15 years | 18 | 14.1 |
| | | From 15 years to less than 20 years | 27 | 21.2 |
| | | From 20 years and over | 9 | 7.08 |

### 4.2. Data Collection

To investigate faculty members' perceptions of the future of distance learning from their educational experience during the COVID-19 pandemic in the KSA, interviews were. They relied on open-ended questions to determine participants' opinions about their perceptions of distance education during the COVID-19 pandemic. Follow-up questions were also asked where appropriate to elicit clarification on the participants' responses. A total of 127 interviews were conducted, taking an average of half an hour per teacher. To protect participant confidentiality while presenting the research data, the participating teachers were coded as P1, P2, P3, . . . , P20 following the order of the data obtained from the interviews.

Research data was collected through participants' written responses to the "standardized open interview", which consisted of a unified series of pre-prepared, sequential questions. Asking all participants in a systematic order provides an advantage in reducing the influence in reductive judgments. The interview form consists of two parts. In the first part, demographic information about the participants is included, and in the second part, open-ended questions about the participants' perspectives on future distance learning concerning teaching during the pandemic are asked.

All procedures performed in the study followed the ethical standards of the institutional research committee of the scientific Research Dean of Hail University (IRB Log Number: RG-21-064) and are in accordance with the 1964 Helsinki Declaration and its later amendments.

### 4.3. Data Analysis

Interviews were transcribed, and data analysis was supported by NVivo12 following a thematic analysis approach. Data collection and analysis took place concurrently. The data were analyzed using inductive analysis. Each participant's answers, especially in the first phase, were coded using keywords so as not to overlap. The data has been entered into the NVivo software with specific codes. Thematic maps show that concepts are organized according to different levels, and potential interactions between concepts are then developed. The analysis team then discussed all the codes and classifications, as well as the possibility of integration between the codes so that the codes could be simplified. This inductive technique allowed the identification of topics presented by participants in response to research questions. The results were presented utilizing a qualitative methodology in accordance with the study's goals.

The analysis began with the creation of a new project in NVivo, dubbed "Future perceptions of faculty staff", and 15 resource volumes within the mentioned project in Microsoft Word format. The next step was encoding, which is putting together excerpts linked together in pools (nodes). The necessary tree nodes were then created.

Based on the thematic analysis, the impact of online teaching on future learning can be characterized by main themes: (a) the importance of training staff on e-learning, (b) an emphasis on blended learning, (c) the development of the technological infrastructure of educational institutions, (d) perceptions of staff for future e-assessment, (e) the most important means of evaluating the performance for the future, (f) applying quality assurance standards in the design and production of e-courses, (g) sourcing reliable resources for the future learning environment, (h) responsibility for producing digital learning resources, (i) providing support and logistics services extensively for digital learning, strengthening the link between the university and the family, and (j) suggested future curriculum.

Congruency was utilized to assure the validity of the qualitative data by selecting expert viewpoints to review and criticize the research processes and analyze its data and reach outcomes comparable to those obtained, which aids in achieving stability in the qualitative data (23).

## 5. Findings and Discussion

### 5.1. Results

To answer the research question, 189 staff interviews were conducted about future perceptions of teaching online based on their experience during COVID-19 and analyzed afterward. The future perceptions were represented by 10 symbols, as shown in Table 2 and Figure 1.

**Table 2.** Themes and Sub-themes Related to the Process of Distance Education.

| The Question | Themes | Subthemes |
|---|---|---|
| Faculty members' perceptions of the future of distance learning | The importance of training faculty members to use E-learning | - Qualifying faculty members, before and during service, in the fields of distance education. |
| | | - Preparing training courses for all those concerned with education planning in the field of distance education management. |
| | | - Training on methods to motivate the learner in the e-learning environment. |
| | | - Focusing on learning management programs and systems and electronic content. |
| | | - Providing the faculty member with future skills in professional development. |
| | | - Modern teaching skills and methods |
| | | - digital skills |
| | | - Teacher skills in the 21st century |
| | Emphasis on blended learning that combines distance education and face-to-face learning | - Adopting e-learning systems permanently in educational institutions and in parallel with face-to-face education in order to achieve more diversity, effectiveness and interaction |
| | The most important means of evaluating teacher performance in the future in light of the digital revolution | - Evaluating the teacher's work by evaluating its outputs in terms of looking at the students' levels and their familiarity with the lessons. |
| | | - by observation |
| | | - Teacher self-assessment |
| | Provide support and logistics services extensively for digital learning | - Improving the digital technological infrastructure in universities in partnership with civil society institutions and all parties of community and national support and funding. |
| | Provide support and logistics services extensively for digital learning | - Improving the digital technological infrastructure in universities in partnership with civil society institutions and all parties of community and national support and funding. |
| | Developing the technological infrastructure of educational institutions | |
| | The most important learning resources suitable for learning in the future learning environment | - Creating educational platforms that help students with asynchronous learning and modern teaching methods to make the educational material attractive to their attention during the lesson |

**Table 2.** *Cont.*

| The Question | Themes | Subthemes |
|---|---|---|
| | Strengthening the link between the university and the family and raising the awareness of parents | - Strengthening the link between the university and the family in order to achieve joint cooperation in activating distance education, |
| | | - Issuing awareness brochures for teachers and parents on the importance of activating distance education |
| | | - Advising parents and students that the objective of education is not degrees or grade points. rather than emphasizing the development of the student's level of performance and awareness, as well as the development of his personality and life skills |
| | Applying quality assurance standards in the design and production of electronic courses, taking into account the characteristics of learners at each stage | |
| | Perceptions and suggestions of faculty members for electronic evaluation methods in the future | - Effectively employing different types of electronic evaluation (diagnostic—formative—final). |
| | | - Peer evaluation or communication |
| | | - Interactive activities—periodic tests—interactive worksheets—simulated tests. |
| | Responsibility for producing digital learning resources | - Ministry of education |
| | | - Experience Houses (Home experiences) |
| | | - Learning resource center specialist |
| | | - faculty members |
| | What future curriculum should use | - learning by discovery |
| | | - Responsibility for producing digital learning resources |
| | | - Responsibility for producing digital learning resources |
| | | - Responsibility for producing digital learning resources |

The themes were used to organize the concepts in Table 2 and Figure 1, after which the possible interactions between the concepts were developed. The analysis team discussed all codes and classifications, as well as the possibility of combining codes to simplify them as subthemes. This inductive methodology enabled the identification of topics presented by respondents in response to research questions. The axes of the investigation were used to structure the qualitative presentation of the results.

The previously mentioned themes were among the views expressed by the sample.

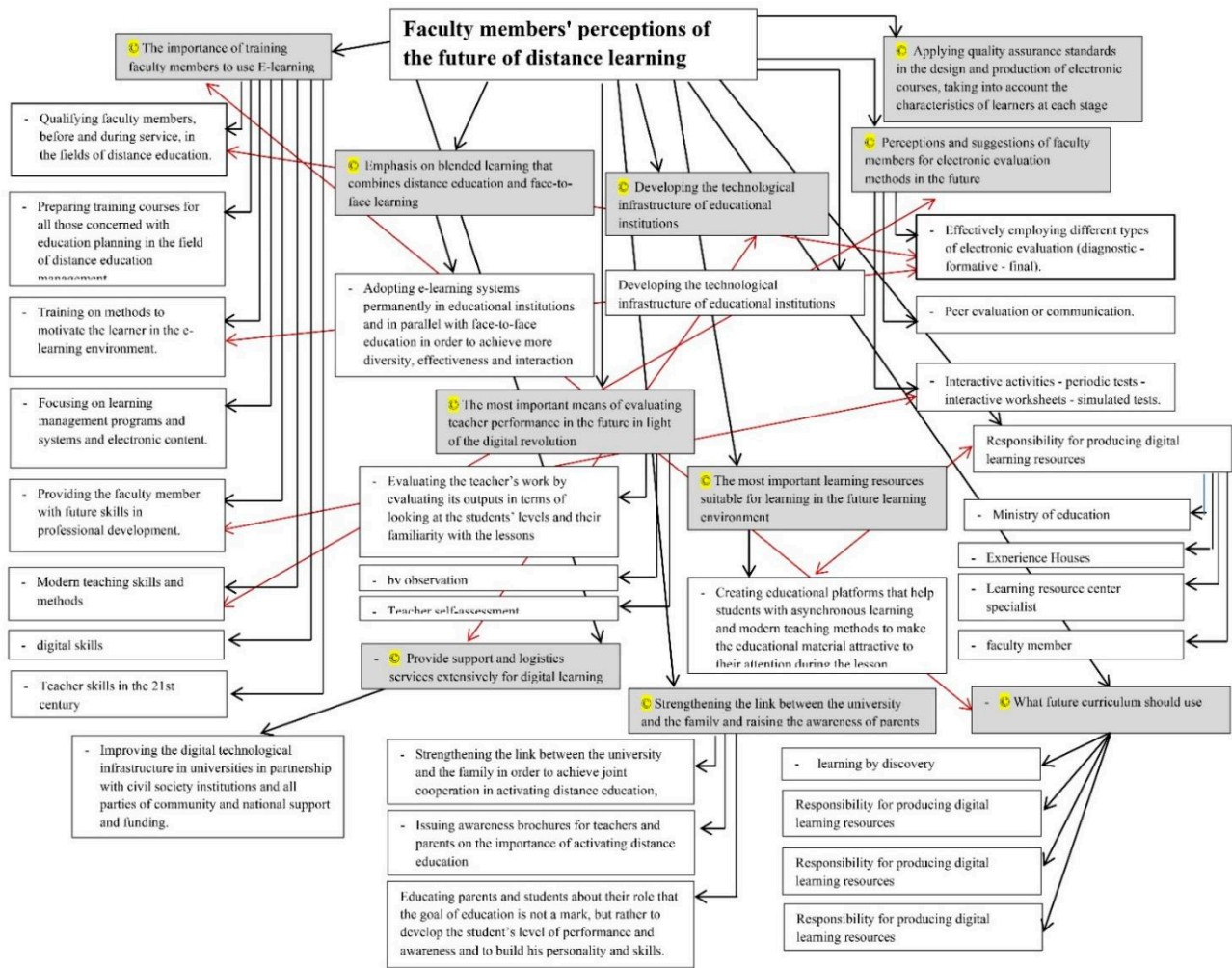

**Figure 1.** The symbols of future perceptions were represented by N Vivo.

5.1.1. Staff s' Future Needs

This global pandemic exposed a significant gap in staff preparation, a crucial factor for emergency remote teaching [8,23–26]. Most of the staff identified the relevance of technology training and strategies, where they shared the following:

- "It is necessary to reconsider staff training with the emergence of remote training to better comprehend its future role in raising awareness and training instructors" (pos. 19).
- "Developing staff's technical skills are reflected in their improved performance, making it easier for them to communicate with students and develop course presentation processes. This enables the staff to smoothly clarify the content of the course, facilitating materials handling" (pos. 38).
- "Post-COVID-19, staff is forced to develop alternative means and use technology in education. It has become important to improve the applications and programs that enable him to enhance their subject presentation" (pos. 72).
- "Familiarity with a 21st Century staff skillset, an emphasis on LMS programs, and familiarity with modern technology tools are necessary technological skills" (pos. 53 & pos. 51).
- "Staff's position hasn't altered much in light of the epidemic, but the challenges have grown. They must be competent to overcome them and efficiently offer educational content" (pos. 22).

New requirements for training educational organization pedagogical staff are linked to developments in educational process organization approaches [24,26,27]. The willingness to embrace change is a major requirement for the successful integration of technology. Yunus (2007); World Bank (2020); and Whalen, (2020) stated that before ICT can be effectively integrated, lecturers should be provided adequate training and support technically, socially, and morally in ICT and pedagogy [8,27,28]. Staff preparation and motivation are key variables in the successful integration of technology in higher education institutions [17,25]. Supporting collaborative types of professional development between teachers, e.g., staff networks, would also be important, as it would allow them to learn from their peers [29].

### 5.1.2. Future Hybrid Learning

Digital learning has become a strategically irreplaceable choice in exceptional circumstances. The findings of Whalen (2020); Marwaha, 2021; and Tondeur. et al.'s (2016) investigation was confirmed [8,13,25]. The following are the perspectives of the sample staff:

- "Hybrid education promotes the efficient diversification of teaching techniques and the development of skills for students" (pos. 21).
- "By integrating distant learning and F2F, education becomes more flexible" (pos. 31).
- "To attain more diversity, effectiveness, and interactivity, e-learning technologies are permanently employed in parallel with F2F education in educational institutions" (pos. 43).

Many educators believe that instructors need more training in using teaching technology, particularly in blended and e-learning. According to a previous study, online learners do marginally better than students in face-to-face environments [24,29,30]. COVID-19 motivated educational institutions to begin investing in LMS to improve their responses to unexpected situations. Blended learning requires a revision of the curriculum, the identifying of the teaching types, and allocating school and home activities [13,31].

The previous results demonstrate the staff's awareness of the growth of technology. With the availability of technology-supported capabilities, a wider scope of guiding methods are available, allowing the advantages of each environment to be realized to the greatest extent possible and realizing that one-to-one teaching is the most effective.

### 5.1.3. Curriculum, Teaching Strategies, and Pedagogy

During the pandemic, many faculty participants changed performance scales. The faculty participants raised several critical points, which we can summarize as follows:

- "Some staff have been struggling with supporting colleagues during the crisis (some staff has created their own support groups inside departments) and how to guarantee students' learning" (pos. 3).
- "Equality and a reduction in student anxiety should be prioritized. Staff should not rely excessively on simultaneous video conferencing to include students with poor internet infrastructure or other family members who need internet bandwidth for other purposes" (pos. 14).
- "It appears that my teaching priorities have shifted. Rather than pondering how to transmit what I've learned, I've decided to convert my courses to a distant learning environment using appropriate electronic teaching methods" (pos. 5).
- "Student-centered design is required for online learning. Focus on content alone will produce poor ineffective multimedia" (pos. 28).

COVID-19 prompted stakeholders in the educational sector to rethink the current teaching approaches. Virtual learning offers flexible learning by providing a learner-based approach [32,33]. The future of higher education institutions will depend on the extent to which they can provide open and adaptable models to diversify teaching/learning processes [34].

- "We can include listening to a podcast, reading a text, or watching a video among the things students should do. This necessitates a thorough analysis of the task, (position 29), i.e., thinking about the practical aspects. (pos. 12)".
- "Online learning relies more on material (readings, videos, exercises, etc.) than on direct in-person interactions (discussions, presentations, etc.), but teachers must source usable good material; on the other hand, it requires students' independence to interact with multimedia (pos. 13)".
- "A precise preemptive design is required for e-teaching (pos. 18)".

Previous mean curricula for remote instruction and broadcast media have not been appropriately created and customized. Therefore, an alternate model can be used when remote instruction is needed. Teachers believe that emergency remote teaching allows them to make a pedagogical shift to a less organized approach to teaching that is more exciting and entertaining [35–37]. Humanizing pedagogy can be operationalized by pushing beyond purely cognitive approaches and becoming more reflexive [38,39]. Teachers are expected to carry out pedagogical innovations to increase student engagement through empathy and care [7,40].

Instead of "predetermined and rigid," the term "curriculum" should be redefined to indicate "adaptable." Schools and teachers should be able to adjust and align their curricula to meet changing societal and individual learning needs. A future curriculum should enable students to learn new skills for the future [12,41].

### 5.1.4. Future Educational Infrastructure, Support, and Logistics

Staff perceptions highlighted the relevance of this dimension in the context of establishing and developing the technical infrastructure of educational institutions. Their thoughts were as follows:

- "It is critical to improve the structure of e-learning and technological infrastructure and equipment in universities (pos. 82)".
- "We must address the internet's weaknesses obstructing future communication with the instructor (pos. 28)".
- "The IoT is used to disseminate knowledge across all aspects of the learning environment (pos. 49 & pos. 15)".
- "Using iPads or smart devices with the students to convert books into interactive e-books with rich multimedia (pos. 19)".
- "Building instructional digital content production centers at universities (pos. 82), providing them with cutting-edge equipment and specialists in digital content production (pos. 34)".

People are increasingly turning to digitalization, and universities will follow suit. These modifications could be long-lasting. COVID-19 highlighted that a robust information technology infrastructure is the key to the success of future education [7,42].

However, the World Bank claims that educational systems are not well equipped to provide online learning on such a massive scale. This is because technological progress frequently outpaces decision-makers ability to keep up in terms of cost and infrastructure support [28,42]. To deliver effective online and blended learning, suitable ICT support in infrastructure and tools must be established. There is no doubt that ICT in academic courses has increased dramatically. Therefore, universities and colleges have begun using LMSs, ARs, CCEs, and educational blogs to enhance pedagogy [30,43]. Governments and education providers should consider equipping faculty and learners with standardized home-based teaching and learning equipment, conducting online professional development, and supporting academic research into online education, especially for students with online learning difficulties [44–46].

Studies show that smart technology will be an important component of learning, especially as the IoT becomes more prevalent in classrooms [13,45,47].

### 5.1.5. Assessment of Learners' Performance

One of the most crucial aspects of remote education is electronic evaluation. It judges the consequences of the skills offered to the student and their understanding. The following is what the participants said in this context:

- "Using current electronic technologies in assessment, such as Microsoft Teams' reading progress, to improve the assessment process (pos. 67)".
- "Activating some features of approved e-learning programs expands the use of educational applications for evaluation (pos. 17)".
- "It is important to use various types of e-assessment effectively (pos. 35) to measure diverse learners' skills (pos. 34)".
- "Assessment tools should be diverse (pos. 87) to increase interest in e-discussions and e-projects, peer evaluation, and the development of more technical assessment instruments (pos. 16)".

Conducting traditional large-scale assessments (written tests/interviews) over the internet was challenging. Online assessments necessitate a complete reassessment (continuous and summative assessments), and staff did their best during the crisis. Assessment is critical since it ensures that learners achieve the course's goals. Various online assessment options should consider challenges.

Rather than proficiency-based abilities, traditional examinations compress knowledge for marks [30,48,49]. During COVID-19, alternative assessments were utilized because of their positive benefits [31,41]. Because courseware platforms evaluate students' abilities at each step, assessing their competencies through questions and answers may become obsolete or inadequate. Many individuals claim that today's assessments are created so that students memorize their material and forget them the following day [15,22]. These forms of assessment can be used to measure authenticity and performance, acting as a relief during this rapid pedagogical transformation.

### 5.1.6. Future Staff Assessment

Staff assessment in distance education improves the profession, thus improving learning outcomes and distance learning quality. The perceptions of the specific category of some of the most essential methods for evaluating future performance are as follows:

- "Teacher assessment approaches that enhance learning and achievement metrics improve retention of and performance of teachers (pos. 42)".
- "It is important to improve staff's proficiency in ICT use, particularly educational applications software, and to be trained for specialization, the extent to which behavioral characteristics and individual differences are activated".
- "A teacher's performance is effectively evaluated to assess student's understanding. Some argue that this is unfair due to some students' poor and unhandled quality—although a minority. Yet, the instructor must raise his students' standards using all available tools (pos. 39)".

Staff perceptions refer to the suggested future evaluation process, ensuring the quality of teaching and remote education [49,50]. The sample determined the responsibility for the evaluation, as shown in Figure 2.

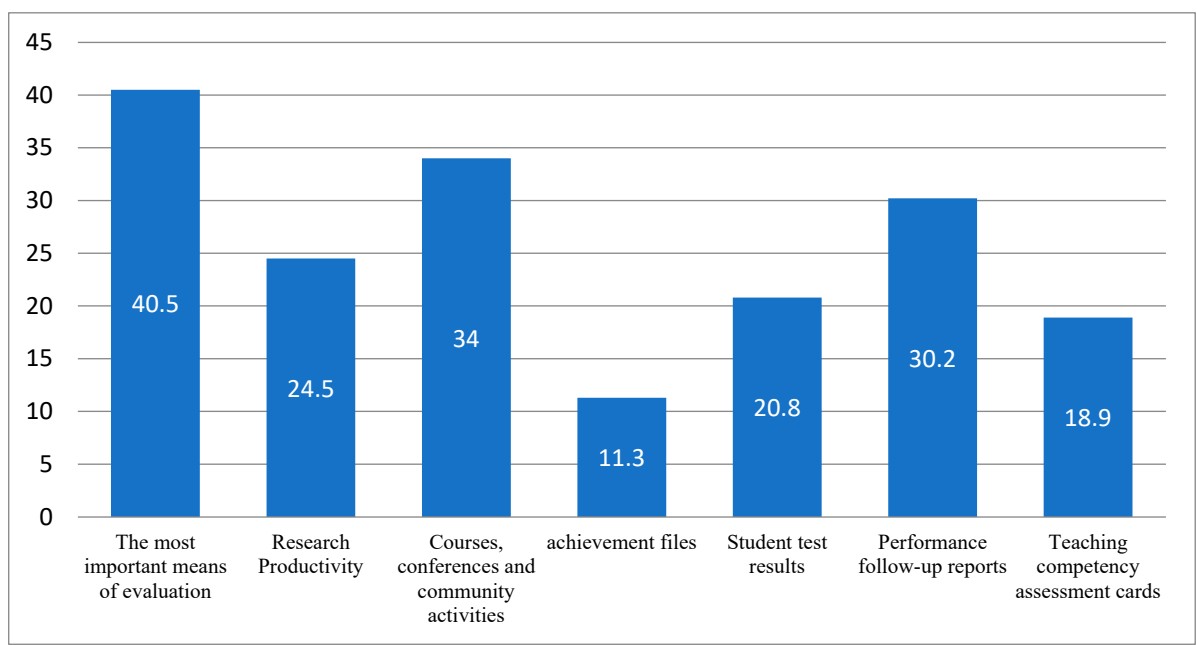

**Figure 2.** Responsible for evaluation.

In-depth studies are needed according to the university regulations in force.

The most important means of evaluating performance in the future were also identified, and are shown in Figure 3.

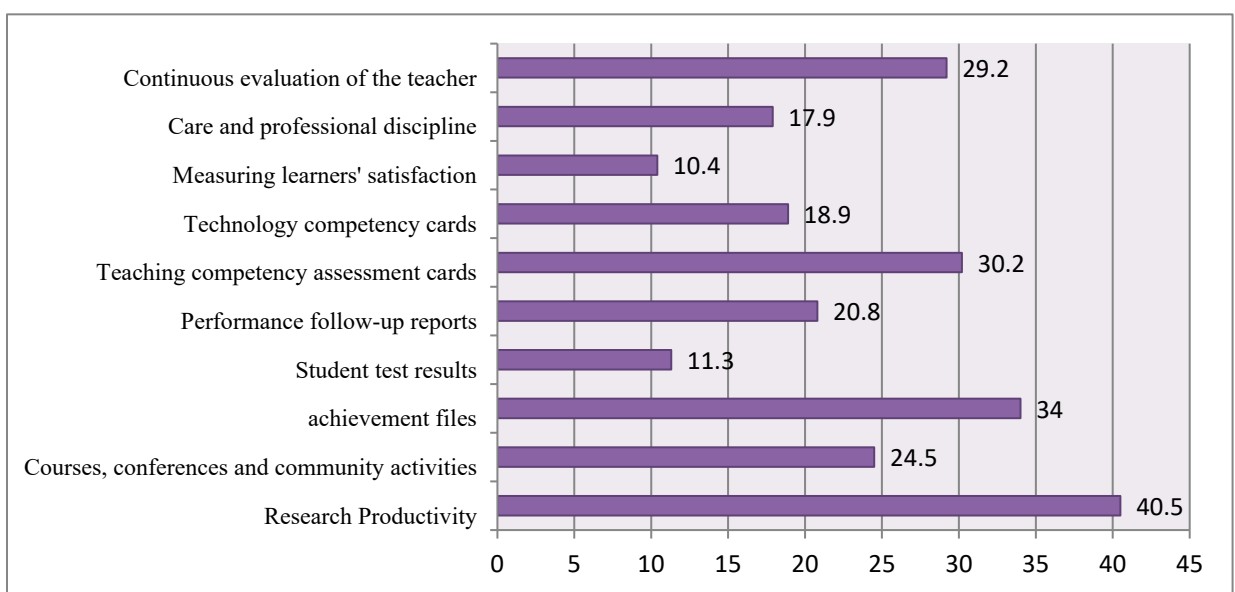

**Figure 3.** The most important method of evaluation.

These means can all be used with relative weight, as indicated by some studies.

5.1.7. Modern Trends in Educational Technology

Recent educational technology trends surged with the emergence of new learning systems, changing how people learn. Participants referred to the following type of remote learning as follows:

- "It is important to source educational platforms that assist students with asynchronous learning and modern teaching methods (pos. 121)".

- "Using tablets, best social communication methods, designing user-friendly websites, localizing appropriate technology from some educationally experienced countries, providing appropriate references for all sciences for free or at nominal prices uploading and making curricula available on appropriate media. The Ministry of Education should offer relevant educational resources for all disciplines, and for the community to participate through institutions and civic society to encourage digital advancement in all educational activities (pos. 129)".
- "The Ministry of Education and private sector firms are responsible for producing the resources (pos. 121)".

  The most desired significant programs were divided into the following categories:

- Unique programs: interactive learning, cyber security, smart programs, interactive multimedia design, montage, and digital transformation.
- Programs that combine personal and technical components, e.g., time management, digital skills and statistics, artificial intelligence, 21st century skills, and cloud computing.
- Purely technical programs, such as design and production abilities for multimedia, and the usage of instructional platforms, infographic design programs, programming, learning analytics, and data mining.

Using new technologies can promote student self-regulation and self-efficacy [37,39,51]. In addition, they can enhance students' quality of life through advancing knowledge and engaging with society [42]. Distance learning–appropriate resources and technology, including videos, discussion forums, social media, etc., were only assessed as effective by a minority of the respondents throughout the remote teaching [3,32].

Governments across the globe are trying to devise open-source e-learning solutions to ensure that barrier-free education includes even the most marginalized students. Future education technology will include asynchronous learning and micro-learning, learning games and simulations, live streaming, virtual reality, augmented reality, robotics, artificial intelligence, data privacy tools, high-resolution videos, blockchain, haptic feedback, and 3D printing [36,39,45].

5.1.8. Parents' Roles in Future Learning

Participants underlined the necessity of improving the distance learning process by enhancing the university-family link and raising parental awareness, as well as extending monitoring and managing students outside the institution to teachers, students, and parents. Participants referred to the most crucial roles of parents:

- "Some parents neglect to follow up on their sons, and we hope that as a result of this problem, parents will be more aware of the need of encouraging and motivating their sons to study" (pos. 25).
- "Some parents may receive a high rating, while others may receive a low rating due to their lack of cooperation and interaction with the university. The cooperation of parents might be invested in future positions where the university collaborates with parents" (pos. 27).
- "Parents can be inspired and encouraged to maximize their future role by paying attention to their children's attendance, completing projects and tests, attending lectures, and following up" (pos. 30).

University closures and home confinement have fundamentally transformed how students, teachers, and parents engage with education [47]. During COVID-19, parents spent more time at home, with the potential to contribute more to children's learning. This pandemic provides a new opportunity for parents to initiate collaboration. Parents from a higher socioeconomic background were more likely to support their children effectively during school closures, either directly, e.g., with emotional support, or indirectly, e.g., with private tutoring [29]. Anger and Heineck found that individuals' cognitive skills are positively associated with their parents', concluding that parental education plays an important role in the transmission of cognitive abilities between generations [52].

Parents must provide a home-based learning environment, social connection, communication, and academic assistance to help children develop their potential, self-esteem, and enhance academic accomplishment [43]. Parents should be aware of the emotional difficulties of online learning. They must learn to help their children emotionally and in their daily home academic tasks.

### 5.2. Discussion

The study aimed to know the staff members' perceptions about online teaching during COVID-19, identify the challenges they faced in adapting to the online learning process, and analyze the staff member's experiences to build future perceptions and identify change drivers in the future learning environment.

Staff members had positive perceptions of the usage of e-learning platforms during COVID-19, although they also emphasized negatives. They emphasized that the shape of future learning settings (in universities) must adapt quickly to accommodate smart technologies. If responsible authorities do not rush to lead change for the future, change will be haphazard.

The impact of virtual learning environments crucially depends on staff's pedagogical and technological readiness and students' and parents' digital competencies. Governments and educational institutions should continue to invest in e-learning. They must carefully analyze all of the lessons learned from this situation.

### 5.3. Limitations

This research shed light on some of the difficulties instructors faced in the classroom during COVID-19. The outcomes of this study should be followed up with research that examines the change base of COVID-19 through time in a variety of higher education systems. This study did not focus on the specificities of each scientific area taught at universities. Future research can explore how each scientific area affects future learning.

### 5.4. Recommendations

The study's findings revealed that staff are not prepared to deal effectively with these shifts in the future. They do, however, demonstrate the need for training as well as fundamental and planned changes to more effectively achieve desired learning objectives. The findings also revealed the potential presented by the COVID-19 pandemic to be as follows:

#### 5.4.1. Staffs' Future Needs

We need interventions to transform teachers from passive users of technology into active designers of technology through training courses for teachers to manage curricula per e-learning quality standards. In addition, it is important to develop their skills in preparing and managing lessons and presentations, e-tests, communication skills, e-activities, guidance, and remote feedback. ICT training must be engaging with learner-centered activities, such as self-directed staff, ongoing practice, conversations with mentors/coaches, and collaboration with colleagues, which would be a helpful way to adapt their practice in the current situation.

It is necessary to introduce courses to prepare the teacher to develop personal learning skills and digital and professional learning.

#### 5.4.2. Future Hybrid Learning

Students will engage with a complex network of both digital and analog texts and spaces. Flexibility is a key feature of future resilient education systems, requiring excellent articulation between levels and types of education and mobilizing alternative modalities of delivery. Future hybrid learning will involve a mix of pedagogies and approaches, as well as the mobilization of alternative pedagogical e-resources from national and worldwide platforms, to provide learners with flexible, smart, adaptive, and quasi-individualized

learning pathways. Stronger ties between formal and non-formal organizations should be established, including the acknowledgment, validation, and accreditation of knowledge and skills obtained through various forms of education.

Professionals and students will choose hybrid industry-ready courses to scale up their abilities to keep up with the latest talents. On the other hand, investments in the technology-based education sector, as well as user needs and preferences (teachers-students-principals-counselors-parents), will grow.

### 5.4.3. For Curriculum, Teaching Strategies, and Curricula

Academic content quality should be maintained, and educational cost leveraging on chances should be limited. Staff at universities need a lot of flexibility concerning education, so they may focus on a human-centered design that encourages problem-solving through empathy building.

Staff must identify digital pedagogy/practices that will aid rather than hinder student learning. It might be simpler to figure out what obstacles students have and address them to create a more inclusive and humanized digital pedagogy. The continuous development of easily accessible open educational platforms is required.

Stimulating curriculum and courses should be modified into e-formats.

This necessitates re-engineering curricula and courses to keep up with smart and networked technologies. Faculties should create, adapt, and harmonize curricula for easier online delivery and management for students to be able to focus. Schedules for teaching, testing, assignments, and other activities should be made that are tolerable for both staff and students.

Courses must depend on various types of virtual learning environments, starting from basic content and scaffolder curriculum-aligned repositories and ending with wide-range synchronous and asynchronous platforms. Curricula should assure equity: social, economic, and technological advancements should benefit all students, not just a privileged few.

### 5.4.4. For Future Educational Infrastructure, Support, and Logistics

Classrooms will exist both physically and virtually to allow students to learn at home while spending class time interacting and applying their knowledge to real-world problems. Access to high-speed internet and to proper ICT tools are basic prerequisites for any future online teaching, necessitating the appropriate technological infrastructure of educational institutions.

### 5.4.5. Assessment of Learners' Performance

Examinations and evaluations should be composed of future learning. This includes relying on electronic evaluation methods and relying on multi-level question banks and evaluation e-portfolios. Also, future assessments can be in the form of evaluations of presentations, interaction models, oral presentations, electronic creative projects, skits or plays, blogpost journaling, one-to-one conferencing, etc.

E-learning solutions backed by smart technologies will make tasks such as evaluation, designing question papers, and preparing grade sheets much easier for teachers. Teachers will be able to focus on activities such as improving teaching quality, developing self-skills, and creating more original course material.

### 5.4.6. For Future Staff Assessment

It is important to build a comprehensive framework for monitoring and evaluating the performance of the smart education system which includes the quality of outputs and their alignment with inputs. The evaluation of the future teacher will depend on a variety of forms.

### 5.4.7. Modern Trends in Educational Technology

Many universities aspire to provide learning experiences that equip students with the skills to compete in the job market and shape the future. Universities must enable students to shape their norms.

Artificial intelligence will endow digital learning with novel ways that are far more engaging than traditional methods. Artificial intelligence allows for the creation of individualized learning experiences that are tuned to meet specific needs. Adopting a tailored learning strategy will assist colleges and universities in resolving common problems. This method will allow educators to keep track of each student's progress.

### 5.4.8. For Parents Roles in Future Learning

Parents will contribute to advisory councils, curriculum committees, and management teams, participating in joint problem-solving at every level, emphasizing the role of the family in the future learning of their children. In addition, parents should establish a learning atmosphere at home that encourages and fosters education and life skills.

### 5.5. Implications of the Study

Pivotal pillars of the future paradigm shift that can be presented concerning the learning outcomes during COVID-19 are:

- The paradigm shift in learning from the traditional site-bound paradigm toward the new CMITriplization paradigm with an emphasis on individualization (human motivation and potential and creativity of the individuals), localization (local resources, community support, and cultural relevance), and globalization (global networking, international support, and world-class resources) in learning
- Building up ecosystems supported by systemic changes in culture, technology, and the paradigm of education for new e-learning through interactive AI technologies, students' self-initiative, and e-learning ecosystems instead of e-text or materials
- The availability of a digital learning objects repository for staff
- A focus on socially connected, learner-centered activities that allow educators to develop knowledge and skills in teaching with technology in any format or situation
- The adoption of unstructured professional development, e.g., mentoring, online forums, or virtual learning groups
- Developing learning theories to develop an effective remote teaching and virtual learning curriculum
- The approval of a set of professional certificates in information and communication technology for teachers based on the Massive Open Online System Courses (MOOCs)
- Building a strong information infrastructure that helps the flow of data between learning networks and employing the IoT and relying on the analysis of Big Data generated by social networking sites for students

The findings are limited to easily quantifiable markers of staff participants, but future research must focus on the more challenging task of studying subtle indications. This includes carefully aligning theory and methodological design to appropriately analyze the phenomenon under examination while contributing to a well-executed body of research in the field of educational technology.

Further research into how future technologies affect cognitive and affective engagement is encouraged. Furthermore, future research is needed to see how lecturers view ICT trading strategies post-COVID-19. We urge stakeholders to seize this once-in-a-lifetime chance to implement fundamental reforms that go beyond improving instructional delivery. The changes we advocate are not novel, but they were never widely adopted in the learning institutions pre-COVID-19. Our most recent experience, however, has increased the need for us to reconsider what is essential and possible for future generations.

**Author Contributions:** Methodology, B.A.A.; Formal analysis, U.M.I., S.M.A. and H.M.D.; Investigation, U.M.I.; Resources, S.M.A. and H.M.D.; Data curation, U.M.I., M.A.M. and H.M.D.; Writing—original draft, S.M.A.; Writing—review & editing, M.A.M.; Supervision, B.A.A. and U.M.I.; Project administration, B.A.A. All authors have read and agreed to the published version of the manuscript.

**Funding:** This research has been funded by Scientific Research Deanship at the UOH, Hail, KSA Through project No. (RG-21016).

**Institutional Review Board Statement:** The research was within the research grants of the Deanship of Scientific Research at the University of Hail, and was applied to students, and did not include Animal experimentation. The methods were performed in accordance with relevant guidelines and regulations and approved by [Deanship of Science at University of Hail].

**Informed Consent Statement:** Informed consent was obtained from all subjects involved in the study.

**Data Availability Statement:** The raw data supporting the conclusion of this article will be available upon request to the corresponding author.

**Conflicts of Interest:** The authors declare that they have no conflict of interest.

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
