# Peer review of "COVID-19 the Gateway for Future Learning: The Impact of Online Teaching on the Future Learning Environment"

_education, doi:10.3390/educsci12120917_

Round 1

Reviewer 1 Report

The need for the study should be added and discussed in detail. During Covid 19, we have seen many related studies. It is better to add literature gap. 

Author Response

COVID-19 the Gateway for Future learning: The Impact of Online Teaching on the Future Learning Environment

The need for the study should be added and discussed in detail. During Covid 19, we have seen many related studies. It is better to add literature gap. 

We reviewed and analyzed studies as best we as a research team could, but were committed to the word count required for publication. In all cases, new references were added according to the needs of the research and the recommendations of the reviewers.

Reviewer 2 Report

This article explores the perceptions of 127 faculty staff in 9 Saudi universities in relation to the future of learning. Although the topic is interesting, I believe there are some aspects that should be acocunted for to improve it.

1- The ideas authors are trying to convey and writing style are not always very clear. One example: In lines 96-98, the authors say "Though not completely novel, student learning spaces will eventually supplant the traditional classroom, making students collaborators in their learning". What are authors trying to say? Isn't the traditional classroom a student learning place where students can collaborate?

2- References do not always justify the arguments the authors try to express. Authors should be more rigorous in the choice of their references. For instance in lines 156 and 157, the studies by Tondeur et al., 2016 and by Bochkareva, 2018 are used to support the following idea "This global pandemic exposed a significant gap in staff preparation, a crucial factor for emergency remote teaching". At the time of those studies, there hadn't been emergency remote teaching.

3- The literature review could include more recent research conducted during COVID19. In relation to challenges and opportunities, this https://link.springer.com/article/10.1007/s10639-020-10258-5 and innovation https://link.springer.com/chapter/10.1007/978-3-030-85469-0_27. Just examples that could be used to expand the first section.

4 - The way data was analysed could be better explained. Who coded the data? Was there a reliability check? How were participants coded and how can we "identify" them along the findings?

5 - Table 1 is difficult to understand. What does the column category refers to? The subtitle of the table is also poor, not explaining very well what one will find in the table.

6 - Figure 1 is very difficult to read, apart from having unedited parts? I suggest a simpler way to present the themes - maybe a table? And what the symbols the authors are referring to? Are they the grey rectangles in the figure? But there is 11, not 10.

7 - I see a poor connection between the themes and the way the results are organized. Also, where does the data for Figure 2 and 3 came from? 

8 - PA style isn't always used correctly (e.g. Basilaia and Kvavadze, 2020 - should be & when in brackets; Burbules, et al, 2020; Burbules, et al. 2020 - al is followed by point and then comma and something wrong with this duplication?; Chelghoum, & Chelghoum, 2020 - the & is not proceeded by comma when citing in-text, etc.).

Author Response

COVID-19 the Gateway for Future learning: The Impact of Online Teaching on the Future Learning Environment

Dear reviewer
I hope you are fine and safe.
First of all, I would like to thank you for your efforts in arbitrating the research and improving its quality. Your recommendations had a great impact and gave more depth.

The notes and action with it: 

  • The ideas authors are trying to convey and writing style are not always very clear. One example: In lines 96-98, the authors say "Though not completely novel, student learning spaces will eventually supplant the traditional classroom, making students collaborators in their learning". What are authors trying to say? Isn't the traditional classroom a student learning place where students can collaborate?

By "learner-centered learning," we mean an approach to education in which students take an active part in determining course content, articulating learning goals, and forming interdisciplinary study groups (which may be different in age, skills, disciplines, and nationalities).

  • References do not always justify the arguments the authors try to express. Authors should be more rigorous in the choice of their references. For instance in lines 156 and 157, the studies by Tondeur et al., 2016 and by Bochkareva, 2018 are used to support the following idea "This global pandemic exposed a significant gap in staff preparation, a crucial factor for emergency remote teaching". At the time of those studies, there hadn't been emergency remote teaching.

A good note.. On the basis of which appropriate references were added to support the idea.

  • The literature review could include more recent research conducted during COVID19. In relation to challenges and opportunities, this https://link.springer.com/article/10.1007/s10639-020-10258-5 and innovation https://link.springer.com/chapter/10.1007/978-3-030-85469-0_27. Just examples that could be used to expand the first section.

A good note.. On the basis of which appropriate references were added to support the idea. A group of references has been deleted, and more than h new references have been added based on the reviewers' notes.

  • The way data was analyzed could be better explained. Who coded the data? Was there a reliability check? How were participants coded, and how can we "identify" them along the findings?

Data Collection and Analysis

To investigate Faculty members' perceptions of the future of distance learning from their experience of education during the Corona pandemic in the Kingdom of Saudi Arabia, interviews were conducted as one of the most widely used data collection methods in social sciences (Yin, 2003). It relied on open-ended questions to reach participants' opinions about their perceptions of distance education during the Corona pandemic. Follow-up questions were also asked where appropriate to elicit clarification on the participants' responses. (106) interviews were conducted, with an average of half an hour/per teacher.

To protect participant confidentiality while presenting the research data, the participating teachers were coded as P1, P2, P3 … P20 following the order of the data obtained from the interviews.

------------------------------------------------------------------------------------------------

The intent of the study was to develop direct interpretations of the data, and then the Explanatory Sequential design strategy was adopted, where the team collected and analyzed qualitative data in order to obtain deeper data. So NVivo software was used.

Validity and Reliability

To ensure the validity of the qualitative data, congruency was used by selecting expert opinions were taken to review and criticize the research steps and analyze its data, and reach results similar to those obtained, which helps in achieving stability in the qualitative research.

  • of the table is also poor, not explaining very well what one will find in the table.

Done as follow:

Table 1. Demographic information for the study sample: University, Academic Rank, gender, and years of experience

No.

Demographic information

n

% of Total

  • Figure 1 is very difficult to read, apart from having unedited parts? I suggest a simpler way to present the themes - maybe a table? And what the symbols the authors are referring to? Are they the grey rectangles in the figure? But there is 11, not 10.

The data were analyzed using inductive analysis (Galloway & Jenkins, 2005). Each participant's answers, especially in the first phase, were coded using keywords so as not to overlap. The data has been entered into the NVivo software with specific codes. Thematic maps show that concepts are organized according to different levels, and potential interactions between concepts are then developed. The analysis team then discussed all the codes and classifications, as well as the possibility of integration between the codes so that the codes could be simplified. This inductive technique allowed the identification of topics presented by participants in response to research questions. The results were presented using the qualitative approach according to the study axes.

  • I see a poor connection between the themes and the way the results are organized. Also, where does the data for Figure 2 and 3 came from? 

The data in figures (2) and (3) come from the analysis of teachers' responses to labnoud shown in the figure.

  • PA style isn't always used correctly (e.g. Basilaia and Kvavadze, 2020 - should be & when in brackets; Burbules, et al, 2020; Burbules, et al. 2020 - al is followed by point and then comma and something wrong with this duplication?; Chelghoum, & Chelghoum, 2020 - the & is not proceeded by comma when citing in-text, etc.).

Done

  • online teaching during the pandemic, while the research question established by the authors refers to the description of future projections regarding teaching. In my opinion, the general objective should be correctly stated.

Done as follow:

This study’s purpose is to study staff members' perceptions of online teaching during COVID-19, describe future projections regarding teaching, and identify the drivers of change in the future learning environment

  • The authors do not explicitly describe objectives or define research variables. I believe that doing so would help to provide expository clarity and to well define the object of the research.

We explain it as follows:

This research aims to collect educators' thoughts on using online learning resources during the recent COVID-19 pandemic. Also, during the COVID-19 pandemic, it's important to examine teachers’ difficulties adjusting to the online teaching and learning process. In addition, for the sake of justification, The question is how effective education was during the deadly Corona epidemic. Investigate how future educators will think about the profession.

  • In my opinion, the literature review and discussion of the results, in comparison with previous works, should be improved. For example, the authors do not refer to previous works that provide very relevant results for their research, such as the following:

https://doi.org/10.3390/educsci12090635

https://doi.org/10.3390/educsci12100688

https://doi.org/10.3390/bs12070203 

https://doi.org/10.3390/educsci12090627

we added it.

  • It is necessary, in my opinion, to increase the depth of the statistical analysis of the quantitative part. To this end, an adequate definition of the research variables will help the authors.

We  do our best.

  • I believe that a Conclusions section should be included, in order to facilitate the reading of the article.

Done.

More changes were made in the corrected file.

All Regards,

Reviewer 3 Report

The article under review is a mixed research on the perception of a sample of Saudi university professors about the impact of the COVID-19 pandemic and the subsequent migration to virtual learning environments on different dimensions of teaching.

There is no doubt that the subject matter is topical and of interest (although the geographical specification of the sample naturally limits the generalizability of the results obtained). However, there are some aspects of the current version of the manuscript that, in my opinion, should be revised by the authors:

1. In the abstract, the main objective of the article is to analyze teachers' perceptions of online teaching during the pandemic, while the research question established by the authors refers to the description of future projections regarding teaching. In my opinion, the general objective should be correctly stated.

2. The authors do not explicitly describe objectives or define research variables. I believe that doing so would help to provide expository clarity and to well define the object of the research.

3. In my opinion, the literature review and discussion of the results, in comparison with previous works, should be improved. For example, the authors do not refer to previous works that provide very relevant results for their research, such as the following:

https://doi.org/10.3390/educsci12090635

https://doi.org/10.3390/educsci12100688

https://doi.org/10.3390/bs12070203 

https://doi.org/10.3390/educsci12090627

4. It is necessary, in my opinion, to increase the depth of the statistical analysis of the quantitative part. To this end, an adequate definition of the research variables will help the authors.

5. I believe that a Conclusions section should be included, in order to facilitate the reading of the article.

Author Response

(The authors gave the same response as above.)

Round 2

Reviewer 2 Report

Dear authors, 

the effort you put into improving your manuscript is relevant. However, I still note crucial aspects that need attention.

Proposed studies were not included and they are certainly current enough to provide your study with a more sustainable background. You tried to use one of the studies, but fail to be rigorous. Reference 49 does not correspond to authors Anger & Heineck (?) and does not conclude any of what you state. A significant point of your work relates to challenges and opportunities to which the study by Lucas & Vicente (2022) is probably one of the most comprehensive to date given the timespan under study. 

APA style continues to be an issue. You opted to use the numerical system, but throughout the text. See for instance line 167.

Regarding that specific paragraph, it does not belong to the results, rather to the section data analysis, as it explains how you analysed data to achieve the topics.

Figure 1 continues to be illegible, contributing to the readers' frustration.

Authors are encouraged to provide a reference for this paragraph "To ensure the validity of the qualitative data, congruency was used by selecting expert opinions were taken to review and criticize the research steps and analyze its data, 157 and reach results similar to those obtained, which helps in achieving stability in the qualitative research." Also, English language should be improved. See different colour.

As to Figures 2 and 3, and because Figure 1 is illegible, its comprehension is very poor. And if you at some point quantified teachers' answers, that should be explained in the data analysis section. Again - the prompt of Figure 2 is Responsible for evaluation and then you have things like "the most important means of evaluation" in the graph. What are you trying to say? And what are the numbers referring to? Percentages? Total number of teachers?

In your answer you say:"The data in figures (2) and (3) come from the analysis of teachers' responses to labnoud shown in the figure." What does labnoud mean?

I believe you can improve your article further and be more rigorous. 

Author Response

Dear Reviewer,

We hope you are doing well and safe. All thanks for your efforts that developing our article. we do our best to make all your recommendations, we attached a file that explains the responses to all recommendations,
All Regards,

Reviewer 3 Report

In my opinion, the authors have adequately satisfied the comments and suggestions made by me and, as a consequence, the quality of the manuscript has been improved. I believe that the revised version of the manuscript presents interesting and well-founded results on the impact of the transition to virtual learning environments resulting from the pandemic. Thanks to the authors for addressing my comments and for their kind response.

Author Response

Dear Editor,

We thank you for your effort in reviewing our article, and your keenness to develop it in a way that benefits scientific research and adds to the field.

Round 3

Reviewer 2 Report

I am willing to accept the paper as long as authors redo Figure 1 - it is clearly a print screen from a PPT slide. Please, unselect the selected elements and ignore the typos identified, and then do a new print screen. Correct graphs or Figures 2 and 3 by correctly identifying percentages.

Author Response

Dear Editor,

I hope you are doing well. we make your recommendations in figure 1 and put too table to explain the results in the figure, also, recommendations in figure 2&3.

Table 1.

Themes and Sub-themes Related to the Process of Distance Education

The question

Themes

subthemes

Faculty members' perceptions of the future of distance learning

The importance of training faculty members to use E-learning

-    Qualifying faculty members, before and during service, in the fields of distance education.

-    Preparing training courses for all those concerned with education planning in the field of distance education management.

-    Training on methods to motivate the learner in the e-learning environment.

-    Focusing on learning management programs and systems and electronic content.

-    Providing the faculty member with future skills in professional development.

-    Modern teaching skills and methods

-    digital skills

-    Teacher skills in the 21st century

Emphasis on blended learning that combines distance education and face-to-face learning

-    Adopting e-learning systems permanently in educational institutions and in parallel with face-to-face education in order to achieve more diversity, effectiveness and interaction

The most important means of evaluating teacher performance in the future in light of the digital revolution

-    Evaluating the teacher’s work by evaluating its outputs in terms of looking at the students’ levels and their familiarity with the lessons.

-    by observation

-    Teacher self-assessment

Provide support and logistics services extensively for digital learning

-    Improving the digital technological infrastructure in universities in partnership with civil society institutions and all parties of community and national support and funding.

Developing the technological infrastructure of educational institutions

-     

The most important learning resources suitable for learning in the future learning environment

-    Creating educational platforms that help students with asynchronous learning and modern teaching methods to make the educational material attractive to their attention during the lesson

Strengthening the link between the university and the family and raising the awareness of parents

-    Strengthening the link between the university and the family in order to achieve joint cooperation in activating distance education,

-    Issuing awareness brochures for teachers and parents on the importance of activating distance education

-    Educating parents and students about their role that the goal of education is not a mark, but rather to develop the student’s level of performance and awareness and to build his personality and skills

Applying quality assurance standards in the design and production of electronic courses, taking into account the characteristics of learners at each stage

-     

Perceptions and suggestions of faculty members for electronic evaluation methods in the future

-    Effectively employing different types of electronic evaluation (diagnostic - formative - final).

-    Peer evaluation or communication

-    Interactive activities - periodic tests - interactive worksheets - simulated tests.

Responsibility for producing digital learning resources

-    Ministry of education

-    Experience Houses

-    Learning resource center specialist

-    faculty members

What future curriculum should use

-    learning by discovery

-    Responsibility for producing digital learning resources

-    Responsibility for producing digital learning resources

-    Responsibility for producing digital learning resources

The previous table shows the organization of the concepts according to the themes, then the possible interactions between the concepts were developed. The analysis team discussed all the codes and classifications, as well as the possibility of integration between the codes so that the codes could be simplified as subthemes. This inductive technique allowed the identification of topics presented by participants in response to research questions

All regards,